# Supplements and Nutritional Interventions to Augment High-Intensity Interval Training Physiological and Performance Adaptations—A Narrative Review

**DOI:** 10.3390/nu12020390

**Published:** 2020-01-31

**Authors:** Scott C. Forbes, Darren G. Candow, Abbie E. Smith-Ryan, Katie R. Hirsch, Michael D. Roberts, Trisha A. VanDusseldorp, Matthew T. Stratton, Mojtaba Kaviani, Jonathan P. Little

**Affiliations:** 1Faculty of Education, Department of Physical Education, Brandon University, Brandon, MB R7A6A9, Canada; 2Faculty of Kinesiology and Health Studies, University of Regina, Regina, SK S4S0A2, Canada; darren.candow@uregina.ca; 3Department of Exercise and Sport Science, University of North Carolina, Chapel Hill, NC 27514, USA; abbsmith@email.unc.edu (A.E.S.-R.); ktrose23@live.unc.edu (K.R.H.); 4School of Kinesiology, Auburn University, Auburn, AL 36849, USA; mdr0024@auburn.edu; 5Department of Exercise Science and Sport Management, Kennesaw State University, Kennesaw, GA 30144, USA; tvanduss@kennesaw.edu (T.A.V.); matthew.stratton@ttu.edu (M.T.S.); 6Faculty of Pure and Applied Science, School of Nutrition and Dietetics, Acadia University, Wolfville, NS B4P2R6, Canada; mojtaba.kaviani@acadiau.ca; 7School of Health and Exercise Sciences, University of British Columbia, Okanagan Campus, Kelowna, BC V1V1V7, Canada; jonathan.little@ubc.ca

**Keywords:** supplements, creatine, caffeine, protein, beta-alanine, glycogen, exercise

## Abstract

High-intensity interval training (HIIT) involves short bursts of intense activity interspersed by periods of low-intensity exercise or rest. HIIT is a viable alternative to traditional continuous moderate-intensity endurance training to enhance maximal oxygen uptake and endurance performance. Combining nutritional strategies with HIIT may result in more favorable outcomes. The purpose of this narrative review is to highlight key dietary interventions that may augment adaptations to HIIT, including creatine monohydrate, caffeine, nitrate, sodium bicarbonate, beta-alanine, protein, and essential amino acids, as well as manipulating carbohydrate availability. Nutrient timing and potential sex differences are also discussed. Overall, sodium bicarbonate and nitrates show promise for enhancing HIIT adaptations and performance. Beta-alanine has the potential to increase training volume and intensity and improve HIIT adaptations. Caffeine and creatine have potential benefits, however, longer-term studies are lacking. Presently, there is a lack of evidence supporting high protein diets to augment HIIT. Low carbohydrate training enhances the upregulation of mitochondrial enzymes, however, there does not seem to be a performance advantage, and a periodized approach may be warranted. Lastly, potential sex differences suggest the need for future research to examine sex-specific nutritional strategies in response to HIIT.

## 1. Introduction

High-intensity interval training (HIIT) involves repeated bursts of vigorous intense exercise (lasting a few seconds up to several minutes) separated by passive rest or low-intensity exercise [1]. HIIT is a viable and time-efficient alternative to induce physiological and cardiorespiratory adaptations [2,3,4]. The specific physiological adaptations induced by HIIT are likely determined by several parameters including intensity, duration, number of intervals performed, the duration and activity during recovery, mode of exercise [5,6], and potentially diet. In untrained and recreationally active individuals, short term (~2 weeks) HIIT is a potent stimulus to induce cardiorespiratory and physiological alterations similar to traditional endurance training, despite lower total exercise volume and training time commitment [7]. Mechanistically, HIIT stimulates peroxisome proliferator-activated receptor-gamma coactivator 1-alpha (PGC-1-alpha) [8], mitochondrial biogenesis, and can enhance whole-body oxidative capacity and maximal oxygen consumption (VO_2_max; [2,7]). Furthermore, HIIT has been shown to enhance peripheral vascular structure and function [9], reduce the rate of glycogen utilization and lactate production (i.e., glycogen sparing) at work matched exercise [6], and enhance lipid oxidation [6]. HIIT has also been shown to improve phosphocreatine recovery kinetics following moderate-intensity exercise [10].

In an athletic population, HIIT is an essential training component to maximize performance and has been used for decades [5]. There is a growing body of evidence suggesting that several nutritional interventions may further augment HIIT by enhancing energy metabolism during exercise thereby increasing total work completed, or by enhancing the adaptive response during recovery leading to an increase in maximal oxygen consumption and exercise performance over time (Table 1). Therefore, the purpose of this review is to highlight recent evidence pertaining to the potential synergistic effects of HIIT and creatine monohydrate, caffeine, nitrate, sodium bicarbonate, beta-alanine, protein, and essential amino acids, as well as in combination with fasting or with a low carbohydrate-based diet, on VO_2_max and exercise performance. Nutrient timing and potential sex-based differences are also discussed. These supplements and dietary strategies for this review were selected based on the potential ergogenic value noted in recent reviews and position stand papers by the International Olympic Committee [11] and the International Society of Sports Nutrition [12], as well as other expert opinions [13].

## 2. Creatine Monohydrate

It is well established that creatine supplementation increases intramuscular creatine stores (Phosphocreatine [PCr] and free creatine; [14,15], which may have a beneficial effect on high-energy phosphate metabolism during periods of intermittent, high-intensity exercise. Theoretically, increased PCr from exogenous creatine supplementation should expand the capacity of the phosphagen energy system and delay the contribution from the glycolytic and oxidative energy systems, leading to greater intermittent, high-intensity exercise performance. PCr hydrolysis consumes hydrogen ions and therefore provides a buffering role against acidosis during exercise [14,16]. Creatine supplementation also increases the shuttling of high-energy phosphate metabolites between the cytosol and mitochondria, leading to increased oxidative recovery [17,18]. Furthermore, creatine increases calcium reuptake into the sarcoplasmic reticulum, which may augment myofibrillar cross-bridge cycling during exercise resulting in greater force development [19,20,21]. Creatine supplementation may likely increase exercise performance and recovery during intermittent bouts of high-intensity exercise. The vast majority of research in this area has focused on cycling performance, with less attention to other exercise modalities such as running, swimming, and skating performance [15]. 

Research is mixed regarding the efficacy of creatine supplementation on intermittent, high-intensity cycling performance. The majority of research suggests that creatine supplementation (5–25 g/day for 3–28 days) can improve total work and average/peak power performance and ventilatory threshold (3–6 cycling trials, 6–120 s of exercise/trial; 20–120 s of rest between trials) in young males and females [22,23,24,25,26,27,28,29,30,31]. However, a few studies have failed to observe the same benefits [32,33,34,35]. Three studies have shown an improvement in intermittent, high-intensity sprint (run) performance (five–six sprints, 15–60 m in length, 30–60 s of rest from creatine supplementation (20 g/day for 5–6 days) in young adults [36,37,38] compared to two studies that failed to observe the same benefits [39,40].

There have been two studies involving creatine supplementation and intermittent, high-intensity skating performance. In elite ice-hockey players, creatine supplementation (20 g/day for 5 days + 5 g/day for 70 days) significantly improved skating performance time across six 80-m skating sprints compared to placebo [41]. In contrast, Cornish et al. [42] failed to observe a beneficial effect from creatine supplementation (0.3 g/kg or ~25 g for 5 days) on skating treadmill performance or blood lactate across repeated 10-s sprints until to fatigue (each sprint was separated by 30-s of rest). 

In regards to intermittent, high-intensity swimming performance, two studies have shown improved performance across three 100-m freestyle sprint swims in junior competitive swimmers (each separated by 60-s of rest) and eight 50-yard swim sprints (each separated by 90-s of rest;) in elite swimmers from creatine supplementation (9–21 g/day for 5–9 days) compared to placebo [43,44]. However, creatine supplementation (20 g/day for 5 days) did not improve swim performance time for either the 25-, 50- or 100-m trial (each separated by a 300 m active recovery swim) or influence blood lactate compared to placebo in highly trained swimmers [45].

Currently, there are only three chronic HIIT studies (>4 weeks), which have examined the potential for creatine to augment training adaptations [23,24,33]. Creatine combined with 4 weeks of HIIT increased ventilatory threshold (VT) [23] and critical power in young males [24] compared to HIIT and placebo. Despite some favorable adaptive responses, there was no effect on whole-body oxygen uptake and time to exhaustion or total work capacity [23,24]. Forbes et al. [33] extended these findings in female participants, demonstrating that 4 weeks of creatine combined with HIIT did not augment improvements in cardiorespiratory fitness, ventilatory threshold, or time trial performance in females when compared to HIIT plus placebo. Potential sex-based differences in response to creatine may explain any differences [46].

While it is difficult to compare results across studies, the inconsistent findings involving different intermittent, high-intensity activities may be related to methodological differences or variables which may influence an individual’s responsiveness to creatine supplementation (for a detailed review, please see [47]. Briefly, creatine supplementation may be more efficacious in those with lower pre-supplementation intramuscular creatine stores [48]. The vast majority of dietary creatine is found in animal-based foods (i.e., meat, seafood, and poultry) and those who consume little to no animal-based products (i.e., vegan, vegetarian) would potentially respond more favorably to creatine supplementation [49]. Furthermore, individuals with the greatest quantity and cross-sectional area of type II muscle fibers appear to respond optimally to creatine supplementation [48]. Regarding sex, some evidence exists that males (not females) experience a decrease in indices of muscle protein breakdown from creatine supplementation [46,50], which may influence recovery from repeated bouts of exercise.

Whether prolonged supplementation with creatine can ultimately augment adaptations to training by increasing training load/intensity or altering metabolic responses to each training bout requires further research. Furthermore, research is warranted to identify variables that would predict the responsiveness to supplementation combined with HIIT.

## 3. Caffeine

Caffeine is one of the most widely studied dietary supplements, with performance benefits dating back to 1978 [51]. Caffeine has consistently demonstrated ergogenic effects on aerobic performance with primary mechanistic advantages acting as an adenosine antagonist to reduce the perception of pain and exertion [52], improving muscle relaxation time by optimizing calcium mobilization in the sarcoplasmic reticulum, and enhancing muscle function by altering the sodium-potassium ATPase pump activity, among others [53]. Mechanistically, there is reasonable support for the use of caffeine in combination with HIIT. Although data describing the effects of caffeine on training is relatively new, collective evidence suggests caffeine may improve sprint performance in trained subjects [54,55,56], but may not be as effective in untrained subjects [57,58,59].

Beyond training status, the type of HIIT may influence the effects of caffeine. Caffeine consumption (3–5 mg/kg body weight) prior to sprint interval training resulted in significantly greater total work completed in the first interval, with maintenance, but not an increase in total work completed in later sprints (bouts four and five) [60]. Similarly, acute and chronic (2 weeks) intake of caffeine (5 mg/kg) prior to sprint interval training yielded no significant improvements in performance or total work but did significantly reduce the appearance of muscle damage markers, supporting the notion that caffeine may reduce exercise-induced muscle damage [61]. Under carbohydrate-restricted or depleted environments, caffeine + carbohydrate (mouth rinse) significantly augmented exercise capacity more than carbohydrate alone or placebo [62,63]. Caffeine alone may directly [64] and indirectly impact HIIT performance. Indirectly, caffeine may support a reduction in muscle damage and inflammation. Acute caffeine (8 mg/kg) may also support a glycogen sparing effect and enhance mitochondrial adaptations and recovery [65].

Contrasting results may be due to responders and non-responders to caffeine supplementation, however, non-responders to caffeine is typically low (~5%) [66]. Responders and non-responders to caffeine may be associated with the CYP1A2 gene, which is known to impact caffeine metabolism [67]. Future research is required to examine the importance of the CYP1A2 gene on caffeine supplementation in conjunction with HIIT in both males and females.

### Multi-Ingredient Caffeine Co-Ingestion

Combinatory effects of caffeine with multi-ingredient supplements have been evaluated prior to participation with high-intensity exercise. Specifically, the majority of pre-workout products on the market include caffeine, along with various doses of other ingredients. To date, there are only a few studies evaluating the effects of these products when used prior to HIIT. Acute ingestion of a multi-ingredient pre-workout containing caffeine, creatine, and amino acids resulted in significant improvements in intermittent running time to exhaustion at 100%, 105%, and 110% of maximal aerobic capacity [68].When the same multi-ingredient supplement was combined with 3-weeks of HIIT (running), there were significant improvements in maximal aerobic capacity (VO_2_max) and lean body mass (+1.2 kg) in active men and women, however, these improvements were not different compared to HIIT and placebo [69].

It has been hypothesized that combining caffeine and creatine may be counterproductive due to competing mechanistic pathways, particularly at high levels of caffeine (i.e., >5 mg/kg; [70]). When evaluated around high-intensity sprint exercise, there does not appear to be an ergolytic effect as consumption of caffeine (5–6 mg/kg) following creatine loading, has been shown to augment high-intensity exercise performance [71,72].

## 4. Sodium Bicarbonate

Increases in hydrogen ion production and subsequent decrease in blood pH levels, known as acidosis, contribute to the onset of peripheral muscular fatigue during high-intensity exercise [73,74,75]. Furthermore, muscle acidosis can inhibit energy production pathways via inhibiting glycolytic enzymatic activity [76], oxidative phosphorylation [76], and regeneration of phosphocreatine [77,78]. Collectively, these mechanisms can impair power output and exercise performance [79,80]. As such, strategies to buffer the exercise-induced intracellular and extracellular acidosis are warranted. One such dietary extracellular buffer is sodium bicarbonate [75]. Sodium bicarbonate is known to increase circulating levels of blood bicarbonate, causing a shift towards a more alkaline environment [81]. This shift in blood pH facilitates an increased rate of hydrogen (H^+^) efflux from the muscle to help maintain muscle acid-base balance by increasing the H^+^ gradient and monocarboxylate transport activity [82,83]. Hence, the efficacy of sodium bicarbonate supplementation is most effective during high-intensity exercise that is largely glycolytic and heavily reliant on the ability to shuttle H^+^ and lactate out of the working muscles [84]. A review from McNaughton and others suggested the benefits of sodium bicarbonate supplementation may be relegated to high-intensity exercise lasting between 1-10 min in duration [85,86].

The overall literature regarding sodium bicarbonate supplementation has shown promise, however, individual studies are mixed. On average, sodium bicarbonate increases performance in acute or repeated bouts of exercise by ~2-3% [86,87]. Such improvements are seen in activities ranging from power sports [88,89], to middle distance events [90], to combat sports [91]. While 2–3% may appear trivial at the elite level, this could certainly influence competition results. For example, in the 2016 Olympic games in track and field, the men’s 800 m the difference between 1st and 6th place was only 1.98%. Zajac and co-authors reported a 2.6% improvement in completion times in a 4 × 50 m swim protocol [92]. Seigler and colleagues similarly demonstrated a 2% increase in completion time utilizing an 8 × 25 m swim protocol [93]. With respect to repeated sprint ability, Bishop et al. [94] demonstrated a 5.1% increase in total work done when utilizing 5 × 6 sec all-out sprints with 30s rest in between bouts.

With the aforementioned improvements in single training sessions, recent investigations have begun to explore chronic supplementation over the length of a full training cycle. Edge et al. [95] showed superior increases in lactate threshold (26% vs. 15%) and time to exhaustion (164% vs. 123%) after 8 weeks of 3x/week training (training progressed from six to 12 two minute intervals at 140–170% of the lactate threshold) in trained female participants who received sodium bicarbonate (0.2 mg/kg at 90 and 30 min) before training compared to placebo [95]. Additionally, Wang et al. [96] demonstrated increases in relative peak power after 6 weeks of HIIT training three times per week in college-aged males [96].

However, it must be noted that not all data support an ergogenic effect of sodium bicarbonate supplementation. In a recent meta-analysis [84], only 18 of the 35 studies reviewed demonstrated improvements in performance. The authors theorized the variance in study outcomes may be partly explained by differences in modalities examined, as well as dosing/loading protocols utilized, and finally, the participant’s circulating bicarbonate response to the supplementation. The majority of previous investigations have used a dosing protocol of 0.2–0.3 g/kg sodium bicarbonate provided 60 to 90 min prior to the exercise bout mixed in 500–2000 mL of water [97,98]. Carr et al. [99] proposed a minimum rise of +5 mmol/L in circulating bicarbonate to elicit an ergogenic effect of sodium bicarbonate supplementation. Jones et al. [100] utilized a dose of 0.1 g/kg and failed to increase any participant’s circulating bicarbonate levels, the requisite 5 mmol/L. In a recent review by Heibel and colleagues [75], only two of the 19 studies that achieved an increase of ≥5 mmol/L failed to see an increase in performance. Moreover, no reviewed study that achieved an increase of ≥6 mmol/L in circulating bicarbonate failed to note an increase in performance.

Despite the potential need to increase sodium bicarbonate dose in order to promote the requisite rise in circulating bicarbonate, doses this high (i.e., ~21 g for a 70 kg athlete) commonly result in gastrointestinal (GI) distress, including nausea, diarrhea, stomach pain, and vomiting [82,87,98,101]. Past investigations and reviews have explored possible solutions to mitigate these common side effects, as it has been noted that many athletes refrain from sodium bicarbonate supplementation due to the aforementioned GI issues [102]. One such meta-analysis by Carr et al. [86] suggested evenly spreading out the desired dosages starting 2–2.5 h prior to exercise in combination with a carbohydrate-rich meal. Additional to the acute loading protocol, a serial loading protocol has also been suggested. In order to utilize the serial method of loading, an individual would consume a slightly higher dosage (0.4–0.5 g/kg/day). However, they would divide the dosage into 3–4 servings throughout the day for 3–5 days prior to exercise [98]. Some evidence suggests circulating bicarbonate levels may be raised for as much as 24 h past the final dosage, and participants may experience less supplement related GI distress [103]. In a recent study, seven (five males, two females) elite middle-distance runners ingested either a placebo, or acutely loading with sodium bicarbonate (0.3 g/kg), or a modified sodium bicarbonate loading protocol (0.6 g/kg over a 19.5-h period) [104]. The modified sodium bicarbonate loading protocol elevated bicarbonate concentration more than the acute loading, without any severe gastrointestinal side effects [104]. However, it must be noted investigations utilizing or exploring this alternate method are sparse, and more studies are certainly needed to determine the efficacy of such modified dosing strategies. Furthermore, it has been recommended by Heibel and colleagues [75] that athletes experiment with multiple protocols and timing to reduce GI symptoms and account for individual responses to sodium bicarbonate supplementation.

## 5. Beta-Alanine

Beta-alanine is a non-proteogenic amino acid produced endogenously in the liver as well as obtained exogenously through the diet from sources such as meat and poultry [105]. Beta-alanine is the rate-limiting substrate in the synthesis of intramuscular carnosine [106,107] that contributes to the buffering capacity of skeletal muscle [105]. Carnosine can also act as an antioxidant [108,109,110]. Potentially, beta-alanine has the ability to enhance training adaptations by increasing the ability to sustain exercise at a higher intensity [111], thus increasing training volume. Beta-alanine may also enhance exercise by altering substrate utilization (i.e., glycogen sparing, reduced blood lactate), reducing oxidative stress [108], as well as increasing the threshold for neuromuscular fatigue [111], and reducing subjective feelings of fatigue [112,113]. Although the precise mechanism(s) remain to be elucidated, beta-alanine with a dose of 3.2 to 6.4 g/day for 4 to 24 weeks has been shown to improve high-intensity exercise performance [105,114], although not all studies are in agreement [115]. Beta-alanine is primarily effective for high-intensity exercise lasting between 1–10 min [105,114], with most data supporting exercise 2–4 min in duration [105,114].

Combining beta-alanine and high-intensity interval training has been shown to improve performance in some studies [116,117], while others have reported no differences compared to placebo and training [118,119,120,121]. Specifically, beta-alanine has been shown to enhance running time trial performance (10 km) following 3 weeks of endurance training [116] and cycling time to exhaustion at 120% peak power following 5 weeks of sprint interval training [117], as well as a trend for greater improvement in cycling time to exhaustion at 110% peak power following 6 weeks of HIIT [122]. In contrast, 4–6 weeks of interval training in recreationally active participants taking beta-alanine was not superior to the same training with placebo supplementation for time trial performance [118,119], power output at ventilatory threshold [120], and critical power during a 3-min all-out cycling test [121].

Despite contrasting results, beta-alanine shows potential for enabling greater HIIT adaptations through increases in training volume [117] and intensity [122]. Contradictory findings may be due to methodological differences that attempted to control for training volume and intensity [119,120,123] or reported no difference [118]. Future research is required to confirm whether performance improvements with beta-alanine supplementation are dependent or independent of training volume.

## 6. Nitrate

The abundant free-radical gas, nitric oxide (NO), is involved in an extensive array of signaling and regulatory processes associated with exercise performance, including blood flow regulation, glucose, and calcium homeostasis, skeletal muscle contractile function, mitochondrial biogenesis, respiration efficiency, and fatigue [124,125,126]. Nitric oxide can be derived from both an oxygen and nitric oxide synthase dependent pathway (e.g., L-arginine NO synthase (NOS) pathway [127,128,129,130]. Therefore, there are two major sources that may lend to NO production; i) end products of the L-arginine NOS pathway, and ii) dietary consumption from nitrate-rich foods, such as green leafy vegetables (e.g., spinach and kale are particularly rich in nitrate) or beetroot juice. To date, several studies indicate that supplementation of dietary inorganic NO_3_^−^ (primarily in the form of beetroot juice) provokes NO production through a series of steps (for review, please refer to [131], and can affect physiological responses to exercise. Interestingly, NO production via NO_3_^−^ supplementation is potentiated as the internal environment becomes more acidic [132], and oxygen tension decreases, creating a hypoxic environment [133]. Given that higher exercise intensity provokes a greater hypoxic and acidic environment within the working skeletal muscle, the use of NO_3_^−^ supplements, such as beetroot juice, for high-intensity exercise may be appropriate. Further, increased blood flow via NO mechanisms has been shown in type II fibers [134,135], as well as improvements in calcium reuptake at the sarcoplasmic reticulum [135]. Such effects could provide performance-enhancing benefits for individuals partaking in intermittent, high-intensity efforts. Initial works conducted on submaximal exercise indicate favorable outcomes, including improvements in efficiency and exercise capacity, mostly for recreational or moderately trained individuals [136,137,138]; however, there are exceptions and elite, well-trained subjects may not respond in the same manner.

Unfortunately, to date, only a few studies have sought to examine the effects of acute dietary NO_3_^−^ on intermittent, high intensity, aerobic-type exercise (e.g., HIIT). This topic has recently been reviewed by Dominguez et al. [139] (no resistance training studies are included in this review). Most studies to date have delivered inorganic dietary NO_3_^−^ in the form of beetroot juice. Of the works that have been completed to date (i.e., eight acute studies; shown in Appendix A), three investigations have reported improvements in performance measures [140,141,142], while four report no changes, and one study, by Martin and colleagues, reported decreased performance [143]. Of the three investigation reporting improvements, participants had a similar, author-defined training status of recreationally trained, as well as the exercise interventions all involved cycling. However, it should be noted that the work: rest ratios and the total volume of exercise was highly variable. Further, the supplementation interventions ranged from acute (180 min prior to exercise) to chronic (7 days of loading) and varied in dosage (8.4–12.9 mmol), making it difficult to provide any sort of direct recommendations or draw definite conclusions. As experimental designs are highly variable between the three studies reporting positive outcomes, a trend for potential improvements in performance, for both submaximal exercise and high-intensity efforts with the implementation of beetroot juice supplementation (i.e., inorganic dietary NO_3_^−^) for lesser trained (e.g., non-elite) individuals continues. However, these conclusions may also be considered controversial, as the training status definition is highly variable within the current literature. For example, training status for participants demonstrating improvements following supplementation included a range of VO_2_peak values (~46–58 mL/kg/min on average); two investigations reporting no effect of dietary nitrate supplementation involved participants with average VO_2_peak values of 49 mL/kg/min and 53 mL/kg/min. Future research should consider more clearly defining the training status of participants. Nonetheless, it appears that higher dosages (8.4 mmol or greater) delivered in the form of beetroot juice may be effective for improving repeated, high-intensity efforts in recreationally trained individuals. However, more research needs to be conducted and it should be noted that the majority of works to date (five of the eight studies report no effect or a decrease in performance) do not support the ergogenic effect of dietary inorganic nitrate for repeated bouts of high-intensity exercise.

Currently, there are several chronic HIIT studies (3–6 weeks) that have examined the potential for nitrate to augment training adaptations [134,141,144,145,146,147]. Two studies found an increase in whole-body oxygen uptake (VO_2_max) following HIIT training (three–four times per week; four–five Wingates per session) in recreationally active participants with nitrate supplementation (12.8 mmol NO_3_^-^ divided into two servings) compared to HIIT and placebo [141,146], while others found no enhancement of aerobic power [134,144,147,148]. Similarly, some studies have shown an enhancement in performance following HIIT with nitrate compared to placebo [141,144,145,146], while others found no benefit [134,148]. Mechanistically, nitrate supplementation seems to have a greater impact on fast-twitch fibers [135] and impact PCG1-alpha expression [149] and, therefore may be ideal for HIIT; however, to date, there appear to be contradictory findings. Future research examining the impact of nitrate on trained athletes with measures of muscle morphology is warranted.

## 7. Protein and Essential Amino Acids 

It is generally accepted that individuals who regularly exercise have greater dietary protein needs [150]. In this regard, dietary protein provides essential amino acids (EAAs) and non-essential amino acids, which act as “building blocks” to sustain, repair, and/or grow intracellular mitochondrial and myofibrillar structures [151]. Elegant tracer work has also established that intramuscular amino acid oxidation can account for approximately 2–5% of ATP regeneration during 60–90-min endurance exercise bouts [152]. Increased protein needs with endurance training were first defined in the 1960s by Gontzea et al. [153] who reported that whole-body nitrogen balance (NBAL) shifted to a negative state in participants who initiated a cardiovascular exercise program daily over a 3-week period while consuming 1.0 g/kg of dietary protein per day. A subsequent study by Tarnopolsky et al. [154] suggested endurance athletes need at least 1.3 g/kg of dietary protein per day to exist in a net positive NBAL [154]. However, there are critiques of the whole-body NBAL method; namely, a general overestimation of nitrogen intake and underestimation of nitrogen excretion, which leads to an underestimation of true protein requirements [155]. The Indicator Amino Acid Oxidation (IAAO) technique is thought to be a more valid method in determining protein requirements given that it measures expired ^13^CO_2,_ which represents the catabolism and oxidation of ^13^C-phenylalanine provided exogenously through the diet [156]. In this regard, recent research by Moore’s laboratory utilizing the IAAO technique suggests that male endurance athletes need 1.7–1.8 g/kg of dietary protein per day [157]. More relevant to the scope of this review, Moore’s laboratory also used the IAAO method to determine that recreational female athletes participating in shuttle running drills need 1.4–1.7 g/kg of dietary protein per day to remain in protein balance [158].

Beyond the increased “protein need” for endurance athletes, there is also select evidence suggesting that a modest enrichment of carbohydrate-laden supplements with protein provides ergogenic benefits during intense endurance exercise. In this regard, Saunders et al. [159] reported that male subjects consuming 1.8 mL/kg body mass of a carbohydrate beverage with added whey protein every 15 min during exhaustive cycling bouts (4:1 ratio, or 26 g carbohydrate and 6.5 g whey protein per 355 mL) experienced 30–40% increases in cycling time to fatigue compared to carbohydrate-only trials. Similar findings have been reported by this research group in males [160] as well as both genders [159]. Interestingly, all three studies also reported that protein-carbohydrate supplementation during exercise significantly reduces post-exercise serum creatine kinase levels relative to carbohydrate-only supplementation suggestive of reduced muscle damage. It is notable, however, that other researchers have reported protein-carbohydrate supplementation during exercise offers no ergogenic advantage or does not prevent post-exercise muscle damage relative to carbohydrate-only supplementation in trained male cyclists [161,162].

Collectively, these studies suggest that HIIT likely increases protein needs above the United States recommended daily allowance (RDA) for adults, which is 0.80 g/kg protein per day. However, there is a large scientific knowledge gap in this area of inquiry, and some caveats for consideration. First, assuming that increases in endurance exercise volume with longer-distance endurance training necessitate the intake of more dietary protein, lower-volume HIIT may not result in the same increase in protein intake needs. Additionally, while 10 weeks of BCAA supplementation (12 g/d) has been shown to increase Wingate peak power and time trial performance in male cyclists [163], there are equivocal data demonstrating that protein supplementation during 6–10 weeks of endurance training fails to promote meaningful improvements in VO_2_peak [164,165], run time performance, cycling time performance [164] or markers suggestive of increased mitochondrial biogenesis [165,166]. While beyond the scope of this review, some of these studies call to question increased protein needs in endurance athletes (e.g., 1.8 g/kg/d) given that participants in non-protein supplemented groups who experienced similar exercise adaptations reported consuming ~1.1–1.3 g/kg protein per day [165,166]. Interestingly, one study determined that young men who consumed 2.4 g/kg of protein per day during a 4-week period with two HIIT cycling sessions per week and three weight training sessions per week gained significantly more lean body mass (LBM, +1.2 kg) and lost more fat mass (FM, -4.8 kg) compared to counterparts consuming 1.2 g/kg protein per day (0.1 kg LBM and -3.5 kg FM) [167]. However, increases in Wingate peak power and VO_2_peak were similar in both groups. Additionally, this study was 4 weeks in length and included participants that were overweight, previously untrained, and consumed a hypocaloric diet designed to induce weight loss.

To date, there is no evidence that suggesting protein intakes higher than RDA would be detrimental to HIIT-induced exercise adaptations. Moreover, some data presented above suggest that in-exercise protein-carbohydrate supplementation or chronic BCAA supplementation could plausibly enhance sprint power or speed-endurance during HIIT sessions. However, data is also lacking, suggesting that individuals engaged in HIIT would experience enhanced exercise adaptations from the purported 1.4–1.8 g/kg/d protein intakes, which are thought to be needed for long-distance endurance athletes. Ultimately, future work is needed in determining if higher-protein diets or protein, EAA, or BCAA supplementation enhances HIIT-induced exercise adaptations, given that there is a paucity of data in this area.

## 8. Carbohydrate Availability and Training Adaptation

An intriguing question in sports nutrition is whether it is a lack or surplus of fuels that facilitate training adaptation [168]. Practically, this relates to whether it is better to provide exogenous fuels (i.e., carbohydrate) before and during training in order to facilitate a greater training load versus strategically restricting carbohydrates before (or during) training in order to promote greater acute metabolic stress that is hypothesized to stimulate greater adaptations. It appears that maximizing carbohydrate availability prior to, and during, HIIT is beneficial to acute performance reviewed by Baker et al. [169], but reducing carbohydrate availability has emerged as a popular strategy in attempts to augment adaptations to HIIT over time. One method of manipulating fuel availability during training is to perform training bouts in the fasted state. Performing moderate-intensity endurance training in the fasted state is well known to result in greater fat mobilization and oxidation during exercise [170,171], and this has been hypothesized to promote greater adaptations in lipid metabolism and mitochondrial enzymes [170]. Whether this leads to improved performance is not yet clear [172]. Hansen et al. [168] popularized the concept of low muscle glycogen training (i.e., “train low”) after showing in a clever, yet small (*N* = 7) study in untrained men who performed single-leg training with low as compared to high muscle glycogen. In order to manipulate training muscle glycogen content, the low muscle glycogen leg trained twice per day with 2 h rest in between bouts every other day whereas the high muscle glycogen leg performed the same volume of training once per day. After 10 weeks, the leg that trained with low muscle glycogen experienced greater improvements in exercise capacity and maximal activity of the key fat oxidation enzyme, 3-hydroxacyl-CoA-dehydrogenase. This study suggested that manipulating carbohydrate availability might be able to stimulate greater endurance training adaptations in skeletal muscle. Other studies have manipulated baseline muscle glycogen content by performing glycogen-depleting exercise (typically a variation of HIIT) on the day prior to a standardized training session with carbohydrate intake restricted during recovery to show that performing a bout of exercise with low muscle glycogen content can augment exercise-induced increases in mitochondrial and metabolic signaling and gene expression [173,174,175]. Follow-up studies using different methods of manipulating carbohydrate availability have examined biochemical and performance adaptations to HIIT, with mixed effects as reviewed by Burke and Hawley and collaborators [176,177,178]. 

### 8.1. Fasted HIIT

There is limited research on the metabolic and performance impacts of performing HIIT in the fasted state. Gillen and colleagues [179] conducted a study in overweight/obese women who performed 6 weeks of interval training (10 X 1-min @ ~90% VO_2_peak) in the fed (training sessions performed 60 min after breakfast) versus fasted (training performed before breakfast) state. Both groups improved body composition and increased markers of mitochondrial capacity in skeletal muscle, with no differences between groups. Terada et al. [180] randomized male cyclists (*N* = 20) to either fasted (no breakfast) or carbohydrate conditions. Participants completed 4 weeks (three sessions per week) of sprint interval training consisting of four to seven, 30 sec all-out sprints with 4 min of active recovery. Training volume and intensity were compromised in the fasted group. However, no differences in changes in VO_2_peak following training and time to exhaustion at 85% of VO_2_peak was longer following training in the fasted group when baseline values were adjusted for. Overall, there is limited research examining the benefits of performing HIIT in a fasted state. Future research in highly trained participants with competition relevant performance outcomes is warranted. 

### 8.2. Low Glycogen HIIT

In contrast to the limited work on fasted HIIT, several studies have examined the acute and chronic responses to low glycogen HIIT. Low glycogen training is typically accomplished by performing an initial glycogen-depleting exercise session (often involving HIIT), withholding or restricting carbohydrate during the post-exercise recovery period, and then performing another training session later the same day or the following morning. Such a strategy is considered a means to manipulate muscle glycogen content, but it must be acknowledged that liver glycogen is likely also lower during the “low glycogen” training session in such protocols. After the initial proof-of-concept “train low” study by Hansen and colleagues [168], other research groups have applied this concept to study how adaptations to HIIT might be influenced by low muscle glycogen. Cochran et al. [181] examined skeletal muscle metabolic responses to HIIT performed under low and high carbohydrate availability using a model whereby participants completed a morning and afternoon training session (5 × 4 min cycling at ~90 VO_2_peak) separated by 3 h of recovery during which subjects ingested a high-carbohydrate drink or non-energetic placebo before the afternoon HIIT trial. Skeletal muscle biopsy samples revealed greater activation of p38 mitogen-activated protein kinase (MAPK) in the low glycogen condition. In a follow-up study using the same carbohydrate manipulation model over two weeks of training [182], low glycogen HIIT (accomplished by performing twice-daily HIIT with the afternoon session performed without carbohydrate intake during recovery) led to greater improvements in 250 kJ time trial performance, yet similar adaptations in mitochondrial enzymes, when compared to performing the same training with high muscle glycogen availability.

A limitation of these studies is that they utilized recreationally active participants, so the results may not be applicable to highly-trained or elite athletes. Yeo and colleagues [183] employed a similar study design in endurance-training cyclist/triathletes whereby one group performed daily training under high muscle glycogen (100-min steady-state ride on one day followed by 8 × 5-min HIIT at maximum self-selected effort the next day) whereas another group trained twice daily with 1–2 h recovery between the steady-state ride and the HIIT session performed every other day. In these highly-trained subjects, only the low glycogen training increased skeletal muscle mitochondrial enzymes and whole-body fat oxidation, yet both groups improved performance. As might be expected, relative power output was lower in the low glycogen group that trained twice daily, leading the authors to conclude that “training low” can lead to greater muscle mitochondrial adaptations and similar performance effects despite compromising high-intensity training capacity. 

Using a different study design to manipulate baseline muscle glycogen content, Bartlett et al. [184] showed that performing running HIIT (6 × 3-min @ ~90% VO_2_max) in low glycogen conditions led to greater activation of signaling pathways related to mitochondrial biogenesis when compared to high glycogen HIIT. In this study, baseline muscle glycogen was manipulated by performing an exhaustive bout of glycogen-depleting exercise on the day before the HIIT running trial in the low glycogen condition only, which resulted in an overall increase in mRNA expression of several mitochondrial marker genes at baseline, making the interpretation of low vs. high glycogen conditions difficult. 

In summary, it does appear that low glycogen HIIT can lead to greater acute increases in some metabolic or stress signaling pathways in skeletal muscle. Whether greater acute metabolic stress/signaling impacts adaptations to chronically performed HIIT under low glycogen conditions is unclear. From a practical perspective, these findings support the modern sport nutrition guidelines advocating a periodized approach to nutrition during training [176,177,185]. Sport nutritionists, coaches, and athletes might want to perform some HIIT sessions under low carbohydrate availability (e.g., off-season, general preparation) in order to try to stimulate greater acute metabolic stress/signaling but it does not appear this approach is superior to training with high or normal muscle glycogen levels over the short term.

## 9. Sex-Based Considerations

There is considerable debate as to whether males and females respond similarly to HIIT. Males have been reported to have greater increases in cardiorespiratory fitness [186], fat loss [187,188], increased mixed muscle protein synthesis [189], and improved metabolic outcomes (e.g., improved glycemic control) [190,191], compared to females. In contrast, one study reported females to have greater improvements in cardiorespiratory fitness [187], while a majority of studies report no effect of sex on cardiorespiratory adaptations [189,190,191,192,193], body composition [187,192,194], or metabolic [187,192,193,195,196,197] effects of HIIT. Despite these suggested differences, very few well-powered studies directly address sex differences in response to HIIT.

Sex differences in response to HIIT have been attributed to males having greater glycogen breakdown during sprints, greater anaerobic capacity, a greater portion of type II fibers, and a greater catecholamine response than women [198]. Metabolically, females rely more heavily on aerobic metabolism during exercise, oxidizing more fat and less carbohydrate than men, attributed primarily to the influence of estrogen [199]. Thus, females are likely to be more fatigue resistant and may recover faster during short rest periods [194].

Due to the divergent sex-based responses in substrate metabolism, there is increasing interest in tailoring nutritional approaches for males and females in order to maximize athletic performance and health. Overall, HIIT appears to be an effective form of exercise for promoting improvements in cardiorespiratory fitness in both men and women; however, future research should investigate how HIIT and nutritional interventions interact in both male and female athletes. 

## 10. Conclusions

With the growing body of evidence and interest surrounding the efficacy of HIIT to enhance mitochondrial capacity and improve exercise performance, and the well-known impact of diet and supplements to alter skeletal muscle metabolism, it is imperative to examine the interaction between HIIT and nutrition. The current narrative review examined several dietary strategies with the potential to augment the benefits of HIIT. Overall, the results show that sodium bicarbonate and beta-alanine show promise for enhancing HIIT adaptations and performance. Beetroot juice/nitrates appear to show some benefits; however, the majority of research suggests limited effectiveness at altering training adaptations, but future research is required to be more conclusive. Caffeine and creatine have the potential to augment HIIT, however, longer-term training studies are lacking. There is a lack of evidence to suggest that a high protein/EAA diet is required to enhance HIIT. With regards to carbohydrate availability, training in a low glycogen state may alter metabolic stress and signaling pathways; however, there does not seem to be a clear performance advantage, and a periodized approach to carbohydrate intake seems warranted. Lastly, the potential sex differences in responses to HIIT and the application of sex-specific nutritional strategies suggest that future research should investigate how HIIT and nutritional interventions interact in both males and females.

## Figures and Tables

**Table 1 nutrients-12-00390-t001:** List of potential dietary supplements to augment acute and chronic high-intensity interval training.

Dietary Supplement/Strategy	Possible Mechanism(s) of Action	Potential Acute Benefits	Potential Chronic benefits	Recommended Dosing Protocol
Creatine	↑ PCr; ↑ Glycogen; ↑ buffering capacity; ↑ calcium handling; Anti-oxidant (?); ↓muscle protein breakdown; ↓ inflammation	↑ high-intensity capacity; ↑ recovery between bouts	↑ training volume (?); ↑ VT and critical power males. 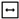 VO_2_max and time trial performance active females	Loading phase: 20 g/day; Maintenance phase: 5 g/day; relative dose: 0.1 g/kg/day
Caffeine	Adenosine receptor antagonist; ↓ perception of pain and exertion; ↑ calcium handling and muscle relaxation; ↓ muscle damage and inflammation (?); ↑ glycogen re-synthesis	↑ total work; ↑ peak power and force production; ↑ muscular endurance; trained > untrained	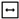 performance, ↓ muscle damage	3–6 mg/kg 45–60 min before exercise.
Sodium Bicarbonate	↓ metabolic acidosis	↑repeated sprints; ↑ high-intensity exercise	↑ lactate threshold; ↑ time to exhaustion; ↑ peak power	Acute loading: 0.2–0.3 g/kg 1–3 h before exercise (>5 mmol/L increase in circulating bicarbonate)Serial loading: 0.4–0.5 g/kg/day divided into three–four servings throughout the day
Beta-Alanine	↑ intramuscular carnosine and buffering capacity; ↓ glycogen utilization; ↓ oxidative stress; ↑ threshold for neuromuscular fatigue; ↓ feelings of fatigue	↑ high-intensity exercise (2–4 min in duration)	↑performance (?); 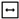 on VT or critical power	3.2 to 6.4 g/day for 2–6 weeks
Nitrate	↑ nitric oxide; ↑ blood flow; ↑ muscular contractility; ↓ O_2_ cost during exercise; muscle fiber type remodeling	↑ high-intensity exercise; ↑ mitochondrial biogenesis	↑ performance (?); ↑ VO_2_max	8–13 mmol of nitrate/day 2–2.5 h before exercise
Protein and amino acids	↑ recovery; ↑ muscle protein kinetics; supports ↑ mitochondrial biogenesis	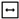 high-intensity exercise; ↑ mitochondrial biogenesis (?)	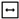 performance or VO_2_max, may benefit the immune system	1.2–2.2 g/kg/day

Abbreviations: ↑ increase; ↓ decrease; ↔ no effect; VT = ventilatory threshold, PCr = Phosphocreatine, VO_2_max maximal oxygen consumption.

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
