# Peer review of "Supplements and Nutritional Interventions to Augment High-Intensity Interval Training Physiological and Performance Adaptations—A Narrative Review"

_nutrients, 2020, doi:10.3390/nu12020390_

Round 1

Reviewer 1 Report

Overall, the paper is good, but in order to be great (which I know this group of authors to be very capable of), I think it needs some revision. The inclusion criteria/focus of the review can be narrowed so the discussion could be more focused in some areas, while some other areas can be removed. 

For example, the title uses the phrase high intensity interval training, which you define as “ repeated bursts of vigorous intense exercise … separated by passive rest or low intensity exercise”, but in some places (177-184, for example) you talk about single efforts like a 70 s all out sprint. This seems off-topic (though I understand sometimes other studies are needed to support mechanistic reasoning, etc).

I think the manuscript would be stronger if the topic were narrowed to HIIT performance (or something similar), which would allow more in-depth discussion. The nitrate section only looks at acute studies (ignoring training studies) while other sections discuss both acute and chronic interventions. Covering acute and chronic studies for all of these areas is too much for a single paper, in my opinion.

There was a recent review that came out in Sports Med looking at a very similar topic but focused on training studies. Of course that doesn’t mean they have a monopoly on the topic but it might be better for this manuscript to focus in more detail on the response to high-intensity interval training performance per se? 

https://link.springer.com/article/10.1007%2Fs40279-019-01185-8

Is the focus on endurance athletes doing HIIT for any type of athlete (including team sports?)

The use of tables and/or figures would be nice. Not necessarily a list of all included studies, but perhaps something about mechanisms or a way to visualize some of the data?

More specifically - 

47 - should it say “reduce the rate of…”

55 - is instead of was?

68 - why are there brackets and parentheses?

68/69 - perhaps ref 12 should be the original work and not a position stand?

99-105 - While I am aware that authors of this review were performing these studies, I feel that a couple of findings relative to this topic of HIIT are missing and warrant at least brief discussion… Both studies (17,18) showing favorable aerobic adaptations (VT, CP) failed to show any differences in high-intensity exercise (TTE at VO2peak and anaerobic working capacity). Based on the justification of creatine in the first paragraph of this section, it might be expected that HIIT performance would be what would benefit.

132/133 - Not sure how you can say “Other recent animal data suggests a potential synergistic effect of caffeine with HIIT…”? The references you cite (52,53) show that caffeine has an antagonistic effect on the changes seen from HIIT. 

213-229 - This study may be worth including, as it’s relevant and offers a promising alternative https://jissn.biomedcentral.com/articles/10.1186/s12970-019-0309-4

234 - act (not acts)

271 -delete the space before the period 

271-274 - this sentence could read more clearly, perhaps split it into two sentences?

257 - this section doesn’t talk about any chronic training studies (of which there are at least 7 by my count). Perhaps that is okay based on my overall feeling that the review could be focused more on acute performance.

284 - “All studies to date have delivered inorganic dietary NO3− in the form of beetroot juice” What about this study https://www.ncbi.nlm.nih.gov/pubmed/29494294 “Discrete physiological effects of beetroot juice and potassium nitrate supplementation following 4-wk sprint interval training.” or this one (again okay to ignore if you focus on acute and not training studies) https://www.frontiersin.org/articles/10.3389/fphys.2016.00233/full

331-335 - should have a reference

336 - add a period after the 144 reference

308 - I’m not sure what the protein section adds, perhaps delete this section to allow space for discussion of other areas?

380 - While this statement is true, I’m not sure this is the best reference as the way most people perform fasted training is after an overnight fast while the reference had people perform a 23 h fast before exercise.

395-397 - “Follow-up studies using different methods of…” - why do you cite three reviews here instead of the actual follow up studies you refer to?

373-397 - some mention of the differences in changes in liver vs muscle glycogen content after an overnight fast is warranted. The first part of this paragraph talks about fasted training and the second part talks about low-glycogen training. The metabolic signals they each lead to may be slightly different.

398 - this paragraph only mentions one training study, but misses another  https://www.ncbi.nlm.nih.gov/pubmed/29619796

However, as mentioned I think a focus on the acute effects of nutrition on HIIT performance would be a better focus for this review. A few studies that come to mind looking at the acute effects of nutrition/fasting on HIIT performance include 

Coffey et al 2011

Pritchett et al 2008

Galloway et al 2014

Little et al 2010

Sherman et al 1989

Taylor et al 2013

405-406 - “there is limited ability or rationale for fasted HIIT to augment training-induced improvements in fat oxidation and mitochondrial biogenesis.” - what about the greater cellular stress that can be induced via the energy-sensing pathways?  (As you mention in subsequent paragraphs) Perhaps it would depend on the volume of the HIIT?

448-452 - Why would people want to try this if it is not superior? I think I understand what you’re saying (perhaps it may be helpful in the long run?) but it could be worded differently.

455-457 - check the punctuation - missing a parentheses. Should there be a reference after fat loss?

480 - “Beetroot juice/nitrates also appear to be effective at altering training adaptations” - you didn’t discuss any training adaptations in that section?

Author Response

The authors would like to thank this reviewer for their feedback and comments. We have provide a response in bold below each comment/suggestion below.  

Comments and Suggestions for Authors

Overall, the paper is good, but in order to be great (which I know this group of authors to be very capable of), I think it needs some revision. The inclusion criteria/focus of the review can be narrowed so the discussion could be more focused in some areas, while some other areas can be removed. 

We thank the reviewer for their thoughtful comments.

For example, the title uses the phrase high intensity interval training, which you define as “ repeated bursts of vigorous intense exercise … separated by passive rest or low intensity exercise”, but in some places (177-184, for example) you talk about single efforts like a 70 s all out sprint. This seems off-topic (though I understand sometimes other studies are needed to support mechanistic reasoning, etc).

We agree with the reviewer and have kept our focus on “interval” training (i.e. repeated bouts).  As such, we have removed discussion of the impact of a single burst of high intensity exercise.

I think the manuscript would be stronger if the topic were narrowed to HIIT performance (or something similar), which would allow more in-depth discussion. The nitrate section only looks at acute studies (ignoring training studies) while other sections discuss both acute and chronic interventions. Covering acute and chronic studies for all of these areas is too much for a single paper, in my opinion.

We do agree that our review is comprehensive in nature. In each section we describe the potential mechanisms, the acute impact on performance, and the chronic effects on performance. We feel that this allows for an overall evaluation of a particular supplement.   

There was a recent review that came out in Sports Med looking at a very similar topic but focused on training studies. Of course that doesn’t mean they have a monopoly on the topic but it might be better for this manuscript to focus in more detail on the response to high-intensity interval training performance per se? 

Thanks for bringing this paper to our attention. We agree that there are similarities between our papers. Our hope is that our review is novel due to being specific to enhancing high intensity interval training and we have discussed other strategies that our not discussed in their review, such as manipulating carbohydrate availability, protein and amino acids, and highlighting potential sex differences.

https://link.springer.com/article/10.1007%2Fs40279-019-01185-8

 Is the focus on endurance athletes doing HIIT for any type of athlete (including team sports?)

We attempted to focus on HIIT specific studies. This would be relevant for both endurance based athletes and team sport athletes; however, based on the current purpose of the review we did not include team sport studies, we feel that this would distract from the current purpose.

The use of tables and/or figures would be nice. Not necessarily a list of all included studies, but perhaps something about mechanisms or a way to visualize some of the data?

We have added a table that summarizes the review and includes potential mechanisms. See table 1.

More specifically - 

47 - should it say “reduce the rate of…”

We have modified as suggested.

55 - is instead of was?

We have corrected this.

68 - why are there brackets and parentheses?

Thanks for catching this, we have modified as suggested.

68/69 - perhaps ref 12 should be the original work and not a position stand?

As suggested we have cited original work.

99-105 - While I am aware that authors of this review were performing these studies, I feel that a couple of findings relative to this topic of HIIT are missing and warrant at least brief discussion… Both studies (17,18) showing favorable aerobic adaptations (VT, CP) failed to show any differences in high-intensity exercise (TTE at VO2peak and anaerobic working capacity). Based on the justification of creatine in the first paragraph of this section, it might be expected that HIIT performance would be what would benefit.

We agree and have updated this section and have clearly highlighted both the negative and positive outcomes of each creatine HIIT study.  

132/133 - Not sure how you can say “Other recent animal data suggests a potential synergistic effect of caffeine with HIIT…”? The references you cite (52,53) show that caffeine has an antagonistic effect on the changes seen from HIIT. 

We have removed this statement.

213-229 - This study may be worth including, as it’s relevant and offers a promising alternative https://jissn.biomedcentral.com/articles/10.1186/s12970-019-0309-4

Thanks for bringing this study to our attention, we have added this into the manuscript.

234 - act (not acts)

We have changed as suggested.

271 -delete the space before the period 

 We have deleted the space.

271-274 - this sentence could read more clearly, perhaps split it into two sentences?

 We have re-organized these sentences to enhance the flow of the paragraph.

257 - this section doesn’t talk about any chronic training studies (of which there are at least 7 by my count). Perhaps that is okay based on my overall feeling that the review could be focused more on acute performance.

We have added in a discussion of these training studies, which is consistent with the other sections.

284 - “All studies to date have delivered inorganic dietary NO3− in the form of beetroot juice” What about this study https://www.ncbi.nlm.nih.gov/pubmed/29494294 “Discrete physiological effects of beetroot juice and potassium nitrate supplementation following 4-wk sprint interval training.” or this one (again okay to ignore if you focus on acute and not training studies) https://www.frontiersin.org/articles/10.3389/fphys.2016.00233/full

 We have modified for clarity.

331-335 - should have a reference

 We have included a reference.

336 - add a period after the 144 reference

 We have added in a period. Thanks for catching this.

308 - I’m not sure what the protein section adds, perhaps delete this section to allow space for discussion of other areas?

Despite limited evidence, there is rationale and potential benefits of supplementing with protein following HIIT training to enhance muscle protein kinetics and recovery. As such, we feel that this section is important and have kept this section in the manuscript. 

380 - While this statement is true, I’m not sure this is the best reference as the way most people perform fasted training is after an overnight fast while the reference had people perform a 23 h fast before exercise.

We have inserted a reference that used overnight fasting.

395-397 - “Follow-up studies using different methods of…” - why do you cite three reviews here instead of the actual follow up studies you refer to?

We have referenced 3 reviews that have discussed the nuances of these methods. We have modified the sentence for clarity.

373-397 (NOW 390-414 in the revised version) - some mention of the differences in changes in liver vs muscle glycogen content after an overnight fast is warranted. The first part of this paragraph talks about fasted training and the second part talks about low-glycogen training. The metabolic signals they each lead to may be slightly different.

We thank the reviewer for this comment and have now mentioned liver and muscle glycogen considerations briefly in this section. We have kept this brief as we are unaware of studies that have been able to directly measure and isolate for the effects of differences in liver and muscle glycogen content. For example, manipulating muscle glycogen by exercise and restricting carbohydrate intake (i.e., “low glycogen training”) inevitably will manipulate liver glycogen (although we are not aware of studies that have directly measured liver glycogen in this context). Likewise, after a 10-12 hour overnight fast (typical duration of fasting for fasted vs. fed exercise studies) there will be slight reductions in both liver and muscle glycogen (for reference, it appears to take ~40-50 hours fasting to fully deplete liver glycogen in humans; Rothman et al. 1991 Science). It does appear that both fasted exercise and the low glycogen training models similarly increase fat oxidation during exercise, as shown in classic fasted exercise studies by Dohm et al. and more recent low glycogen training studies.

398 - this paragraph only mentions one training study, but misses another  https://www.ncbi.nlm.nih.gov/pubmed/29619796

Thank you for bringing this study to our attention. We have added this to the fasted HIIT paragraph.

However, as mentioned I think a focus on the acute effects of nutrition on HIIT performance would be a better focus for this review. A few studies that come to mind looking at the acute effects of nutrition/fasting on HIIT performance include to highlight key dietary interventions that may augment adaptations to HIIT

Coffey et al 2011

Pritchett et al 2008

Galloway et al 2014

Little et al 2010

Sherman et al 1989

Taylor et al 2013

We thank the reviewer for this comment and additional references. As the stated objective of this review is “to highlight key dietary interventions that may augment adaptations to HIIT” we do not feel it is within scope to address all possible nutritional strategies that may impact acute HIIT performance. Such a review would need to encompass all related work on how nutrition impacts intermittent exercise/sport performance and we feel this would substantially detract from the focus of our review, which has the goal of summarizing possible nutritional manipulations that might augment training adaptations. We have referenced acute studies examining molecular signaling in skeletal muscle within the carbohydrate section to highlight the mechanistic basis for why fasted or low glycogen training may stimulate greater training adaptations; this section is not meant to be a review of the impacts on acute performance. However, we have added the following sentence at the start of this section in order to further clarify this point regarding the lack, or surplus, of fuel and training adaptations. The Baker et al. 2015 Nutrients reference is a review article summarizing the impacts of acute carbohydrate supplementation on high-intensity intermittent exercise performance, which encompasses studies of HIIT and many team sport protocols.

"It appears that maximizing carbohydrate availability prior to, and during, HIIT is beneficial to acute performance as reviewed by Baker et al. but reducing carbohydrate availability has emerged as a popular strategy in attempts to augment adaptations to HIIT over time."

405-406 - “there is limited ability or rationale for fasted HIIT to augment training-induced improvements in fat oxidation and mitochondrial biogenesis.” - what about the greater cellular stress that can be induced via the energy-sensing pathways?  (As you mention in subsequent paragraphs) Perhaps it would depend on the volume of the HIIT?

We agree and have modified this sentence for clarification.

448-452 - Why would people want to try this if it is not superior? I think I understand what you’re saying (perhaps it may be helpful in the long run?) but it could be worded differently.

 We have clarified this sentence.

455-457 - check the punctuation - missing a parentheses. Should there be a reference after fat loss?

We have modified as suggested. 

480 - “Beetroot juice/nitrates also appear to be effective at altering training adaptations” - you didn’t discuss any training adaptations in that section?

Now that we have added in the training papers, we have kept this sentence in the conclusion.

Reviewer 2 Report

This manuscript presents a narrative review about high intensity interval training (HIIT) and possible dietary interventions that may augment adaptations to HIIT.

Text and reference format (in the text) are not in accordance with journal requirement; please check and correct it

Line 55-59: Enumerating this list “(1) creatine monohydrate, (2) caffeine, …” would be better without numbers.

Line 73-75: is there a reference for this statement?

Line 110-114. Be consistent: make or no summary for all of mentioned supplements/interventions in these subsections.

Line 140 onward: what kind of supplements are “multi-ingredient supplement”? I.e. in cited study (reference 57).

Line 187: Zajac and others – others or coauthors?

Line 304-307: references are needed

Line 402: subscript is missing in VO2

Line 491-492: why this table is not cited in the text? And what about the references included in the table? Are all of them cited in reference section?

There is need to explain why this supplements were chosen. Could you refer to the opinions of the AIS, ISSN and ACSM in relation to all of chosen supplements?

The following reference also might be useful: Close et al.: New strategies in sport nutrition to increase exercise performance. Free Radical Biology and Medicine 2016, 98, 144-158.

There are many old references – are all of them necessary and indispensable? More than 40 references are over 15 year old.

I would suggest the title correction: “Supplements and nutritional interventions to augment (..)” because in this study supplements are mostly described.

I would like to encourage the authors to prepare table describing all of mentioned supplements and intervention in relation to possible effect in relation to all of analyzed (mentioned in the text) groups: recreationally active, highly-trained and elite athletes. Data organization is needed – in my opinion table is a good choice for this and will make you manuscript more clearly.

Author Response

First we would like to thank the reviewer for their effort and thoughtful comments. We have responded to each comment/suggestion below.

Comments and Suggestions for Authors

This manuscript presents a narrative review about high intensity interval training (HIIT) and possible dietary interventions that may augment adaptations to HIIT.

Text and reference format (in the text) are not in accordance with journal requirement; please check and correct it.

We have carefully reviewed the manuscript and formatted according to the journal guidelines.

Line 55-59: Enumerating this list “(1) creatine monohydrate, (2) caffeine, …” would be better without numbers.

We have modified as suggested. Please see the revised manuscript.

Line 73-75: is there a reference for this statement?

As suggested, we have inserted a reference for this statement.

Line 110-114. Be consistent: make or no summary for all of mentioned supplements/interventions in these subsections.

We have adjusted each section to ensure consistency throughout. 

Line 140 onward: what kind of supplements are “multi-ingredient supplement”? I.e. in cited study (reference 57).

We have added in specific information to clarify the contents of the multi-ingredient supplement used in reference 57.

Line 187: Zajac and others – others or coauthors?

Thanks for catching this error, we have modified as suggested.

Line 304-307: references are needed

We have inserted the references and specific details of this section are shown in the supplemental table, which we have now referred to in the text.

Line 402: subscript is missing in VO2

Thanks for catching this error, we have modified as suggested. 

Line 491-492: why this table is not cited in the text? And what about the references included in the table? Are all of them cited in reference section?

I have cited this table in the text, and all the references in the table are listed in the reference section.

There is need to explain why this supplements were chosen. Could you refer to the opinions of the AIS, ISSN and ACSM in relation to all of chosen supplements?

We agree that this list is based our expert opinion (hence we stated in the title that this is a narrative review). However, we have also added a statement: “These supplements and dietary strategies for this review were selected based on the potential ergogenic value noted in recent reviews and position stand papers by the International Olympic Committee (11) and the International Society for Sport Nutrition (12), as well as other expert opinions (13).”

The following reference also might be useful: Close et al.: New strategies in sport nutrition to increase exercise performance. Free Radical Biology and Medicine 2016, 98, 144-158.

Thank you for this recommendation, we have inserted this reference into the manuscript.

There are many old references – are all of them necessary and indispensable? More than 40 references are over 15 year old.

For each supplement we have provided some historical background (hence older references). We believe that this research (despite its age) is valuable to the review. We have carefully reviewed the references and made changes if deemed indispensable.

I would suggest the title correction: “Supplements and nutritional interventions to augment (..)” because in this study supplements are mostly described.

We agree and have modified as suggested.

I would like to encourage the authors to prepare table describing all of mentioned supplements and intervention in relation to possible effect in relation to all of analyzed (mentioned in the text) groups: recreationally active, highly-trained and elite athletes. Data organization is needed – in my opinion table is a good choice for this and will make you manuscript more clearly.

We have created a table that summarizes each supplement and the potential mechanisms. We hope that this is sufficient and adds value/clarity to the paper. We have also clarified whether participants were recreationally active, highly trained, or elite throughout the manuscript.

Round 2

Reviewer 1 Report

Thank you for your updates.

I still have the same overarching concern, however, and that is a lack of depth in the analysis. For example in the creatine section, the first paragraph provides a nice mechanistic framework, then the next four paragraphs shows contrasting results, followed by a conclusion that says it is difficult to compare results across studies. The contrasting results are fine, but isn’t the whole point of a narrative review to talk about how/why things work in some conditions and not others, and how researchers or practitioners can move forward effectively? Perhaps posing specific research questions that are needed would be helpful?

Some specific comments…

143 - “Caffeine alone may not have large direct implications for HIIT performance” What about Lane et al 2013? (Caffeine Ingestion and Cycling Power Output in a Low or Normal Muscle Glycogen State - “independent of glycogen availability, caffeine enhanced work ca- pacity during intense interval training by approximately 3%.”)

145-146 - perhaps worth mentioning that this study used extremely large amounts of caffeine (8 mg/kg)?

183-184 - Not super important, but 800 m what? swimming? running?

193 - You deleted the prior sentences talking about single bouts, so the next sentence that starts with “Aside from single high-intensity bouts…” seems odd. Should you also delete that next sentence and just start with “Zajac…”?

266-268 - same comments as creatine, “Contradictory findings may be associated with training status (who is it better for?), training intensity (when or how should people use it or what research should be done?)

315-317 - “it should be noted that the majority of works to date (five of the eight studies report no effect or a decrease in performance) do not support the ergogenic effect of dietary inorganic nitrate for repeated bouts of high-intensity exercise.” This seems at odds with your conclusions?

504-506 - should the hyphen be a comma? and then should there be a comma after metabolism?

Author Response

Thank you for your updates.

I still have the same overarching concern, however, and that is a lack of depth in the analysis. For example in the creatine section, the first paragraph provides a nice mechanistic framework, then the next four paragraphs shows contrasting results, followed by a conclusion that says it is difficult to compare results across studies. The contrasting results are fine, but isn’t the whole point of a narrative review to talk about how/why things work in some conditions and not others, and how researchers or practitioners can move forward effectively? Perhaps posing specific research questions that are needed would be helpful?

To address your concern, we have added in discussion pertaining to why contrasting results exist. In particular we have enhanced the creatine section. 

Creatine: "While it is difficult to compare results across studies, the inconsistent findings involving different intermittent, high-intensity activities may be related to methodological differences or variables which may influence an individual’s responsiveness to creatine supplementation (For a detailed review, please see Candow et al. 2019). Briefly, creatine supplementation may be more efficacious in those with lower pre-supplementation intramuscular creatine stores (Syrotuik and Bell, 2003). The vast majority of dietary creatine is found in animal based foods (i.e., meat, seafood, poultry) and those who consume little to no animal-based products (i.e., vegan, vegetarian) would potentially respond more favorably to creatine supplementation (Burke et al., 2003). Furthermore, individuals with the greatest quantity and cross-sectional area of type II muscle fibers appear to respond optimally to creatine supplementation (Syrotuik and Bell, 2003). Regarding sex, some evidence exists that males (not females) experience a decrease in indices of muscle protein breakdown from creatine supplementation (Johannsmeyer et al., 2016; Parise et al. 2001) which may influence recovery from repeated bouts of exercise."

Caffeine section: 

Contrasting results may be due to responders and non-responders to caffeine supplementation, however, non-responders to caffeine is typically low (~5%) [67]. Responders and non-responders to caffeine may be associated with the CYP1A2 gene, which is known to impact caffeine metabolism [68]. Future research is required to examine the importance of the CYP1A2 gene on caffeine supplementation in conjunction with HIIT in both males and females.

Beta-alanine:

Despite contrasting results, beta-alanine shows potential for enabling greater HIIT adaptations through increases in training volume [118] and intensity [123]. Contradictory findings may due to methodological differences that attempted to control for training volume and intensity [120, 121, 124] or reported no difference [119]. Future research is required to confirm whether performance improvements with beta-alanine supplementation are dependent or independent of training volume.

Some specific comments…

143 - “Caffeine alone may not have large direct implications for HIIT performance” What about Lane et al 2013? (Caffeine Ingestion and Cycling Power Output in a Low or Normal Muscle Glycogen State - “independent of glycogen availability, caffeine enhanced work ca- pacity during intense interval training by approximately 3%.”)

We have modified the sentence. "Caffeine alone may directly (Lane et al. 2013) and indirectly impact HIIT performance."

145-146 - perhaps worth mentioning that this study used extremely large amounts of caffeine (8 mg/kg)?

We agree that the dose is important and have added in the 8 mg/kg. 

183-184 - Not super important, but 800 m what? swimming? running?

We have clarried that this was running. 

193 - You deleted the prior sentences talking about single bouts, so the next sentence that starts with “Aside from single high-intensity bouts…” seems odd. Should you also delete that next sentence and just start with “Zajac…”?

We agree and have deleted the sentence "Aside from single high...."

266-268 - same comments as creatine, “Contradictory findings may be associated with training status (who is it better for?), training intensity (when or how should people use it or what research should be done?)

We have updated this section for clarity. "

While it is difficult to compare results across studies, the inconsistent findings involving different intermittent, high-intensity activities may be related to methodological differences or variables which may influence an individual’s responsiveness to creatine supplementation (For a detailed review, please see Candow et al. 2019). Briefly, creatine supplementation may be more efficacious in those with lower pre-supplementation intramuscular creatine stores (Syrotuik and Bell, 2003). The vast majority of dietary creatine is found in animal based foods (i.e., meat, seafood, poultry) and those who consume little to no animal-based products (i.e., vegan, vegetarian) would potentially respond more favorably to creatine supplementation (Burke et al., 2003). Furthermore, individuals with the greatest quantity and cross-sectional area of type II muscle fibers appear to respond optimally to creatine supplementation (Syrotuik and Bell, 2003). Regarding sex, some evidence exists that males (not females) experience a decrease in indices of muscle protein breakdown from creatine supplementation (Johannsmeyer et al., 2016; Parise et al. 2001) which may influence recovery from repeated bouts of exercise.

315-317 - “it should be noted that the majority of works to date (five of the eight studies report no effect or a decrease in performance) do not support the ergogenic effect of dietary inorganic nitrate for repeated bouts of high-intensity exercise.” This seems at odds with your conclusions?

We have adjusted the conclusion to better reflect the data. 

504-506 - should the hyphen be a comma? and then should there be a comma after metabolism?

We have changed as suggested.

Reviewer 2 Report

Dear Authors,

Thank you for addressing my concerns and comments, revising the manuscript and adding Table 1. There has been clarification and your honesty is appreciated. 

Author Response

Thank you very much for your support.